# Informational ecosystems partially explain differences in socioenvironmental conceptual associations between U.S. American racial groups
Roberto Vargas ✉ & Timothy Verstynen

Social groups represent a collective identity defined by a distinct consensus of concepts (e.g., ideas, values, and goals) whose structural relationship varies between groups. Here we set out to measure how a set of inter-concept semantic associations, comprising what we refer to as a concept graph, covaries between established social groups, based on racial identity, and how this effect is mediated by information ecosystems, contextualized as news sources. Group differences among racial identity (278 Black and 294 white Americans) and informational ecosystems (Left- and Right- leaning news sources) are present in subjective judgments of how the meaning of concepts such as healthcare, police, and voting relate to each other. These racial group differences in concept graphs were partially mediated by the bias of news sources that individuals get their information from. This supports the idea of groups being defined by common conceptual semantic relationships that partially arise from shared information ecosystems.

Social groups are defined by a set of individuals who hold a common social identification or view themselves as members of the same social category[1]. According to social identity theory, groups provide a framework for understanding oneself in the context of a larger societal framework[2]. Consequently, group frameworks represent a collectively shared identity, defined not only by a common set of ideas, values, and goals, but also by a collective semantic knowledge that serves as a lens through which the environment is given meaning. This collective knowledge provides a tool for reducing individual bias and increasing group accuracy for problem-solving and decision-making (i.e., "wisdom of the crowds"[3]); however, in some contexts, collective knowledge can also introduce heuristic biases reinforcing stereotypic associations towards specific social identity groups[4], reflected in the "bias of the crowds" theory[5].

If groups are defined by a shared set of concept semantic relationships, then how does this collective concept map develop? Early social theories emphasized that knowledge is the product of social interactions[6]. This constructionist framework proposed that a group's unified understanding of reality is shaped primarily by social interactions and agreements on the meanings assigned to common experiences, where shared practices and negotiations defined collective understanding that evolved with interactions within a community[7]. However, another source of collective meaning comes from shared information ecosystems (e.g., television, social media, websites). It is theorized that news sources, through repeated coverage of issues surrounding specific concepts, communicate not only that concept schemas or frameworks are shared by a group of individuals, but also provide specific criteria for how concepts should be subsequently evaluated[8,9]. Selective consumption of media messaging may prioritize the importance of certain topics over others and frame the perspectives people use to evaluate an issue or topic (e.g., viewing immigration as an economic threat or moral issue[8]. In extreme cases, selective media consumption may enhance beliefs in conspiratorial viewpoints[10]. By continuously raising awareness about an individual or issue, media can sculpt concept relations by reinforcing or severing associative information. For example, on March 8th of 2020 there was a 650% increase in the use of the term "Chinese virus" on Twitter (now X) following the use of stigmatizing language by media outlets on March 8th[11], reflecting a suddenly strengthened association between the concepts of Chinese identity and the COVID-19 pandemic. Taken together, this suggests that information ecosystems may influence the collective relationship of concepts shared by members within a given social group.

In the United States, one of the most distinct group boundaries is racial identity. As groups, white and Black identifying Americans often have different attitudes and beliefs on issues like law enforcement[12], healthcare[13–15], religion[16], and political leaning[17]. At the cognitive and

Department of Psychology, Carnegie Mellon University, Pittsburgh, PA, USA. ✉e-mail: robertov@andrew.cmu.edu

neural levels, the strength of the association between unique concepts can be mapped out as a graph, defining the unique "concept geometry" of representations for each person[18–24]. Associative relationships among concepts and attitudes have the potential to provide insight into a deeper meaning structure when measured between social groups, potentially informing a larger cultural context[25]. If shared information ecosystems (e.g., news sources) contribute to how groups build their collective concept graphs, then we should see at least part of the variability in group-level concept graphs being mediated by the biases of their information sources. That is, to what extent are group differences in concept associations attributable to related information consumption rather than possibly lived experiences? Here we tested this idea by exploring the relationship between various facets of identity (including race, gender, income, and age) and the political bias of the news subjects consumed. This relationship between identity and news bias can be understood within the architecture of conceptual associations, referred to as a concept graph, consisting of socioenvironmental concepts (*police, firefighters, healthcare, science, religion, voting, immigration, neighbors, liberals*, and *conservatives*) and attitude concepts reflecting 'primitive' emotions (*anger, sadness, fear, joy, love, trust*)[26]. Specifically, we hypothesize that (a) there will be reliable differences in concept graphs between Black and white identifying participants for socioenvironmental concepts; (b) that informational ecosystems, represented as the bias in the news participants consume, are associated with differences in concept graphs; and (c) that informational ecosystems will at least partially mediate (i.e., partially explain) identity-based differences in concept graphs.

## Methods

### Participants

A sample of 600 participants were recruited via Amazon Mechanical Turk (MTurk; using Cloud Research MTurk Toolkit) and Cloud Research Connect. Participants were included in the sample if they completed the study session (585), identified as non-Hispanic white American or Black/African American (572), and identified as men or women (567). We did not differentiate between cisgender or transgender participants. For news-based analyses, only participants who included news consumption responses which could be assessed for bias were included. This restriction resulted in a final sample of 446 participants (for complete demographic breakdown see Supplementary Fig. 1). Due to technical complications with MTurk, recruitment was changed from MTurk to Connect midway through data collection. There is a chance the same participant could have been recruited from both platforms, however, to catch if this had occurred, participant's MTurk identifiers were queried during their recruitment via Connect. MTurk identifiers from participants recruited through Connect were then cross referenced with those previously collected from MTurk. No repeats were identified. Informed consent was obtained from all participants in accordance with the study protocol as approved by Institutional Review Board of Carnegie Mellon University. No study materials were preregistered for this study.

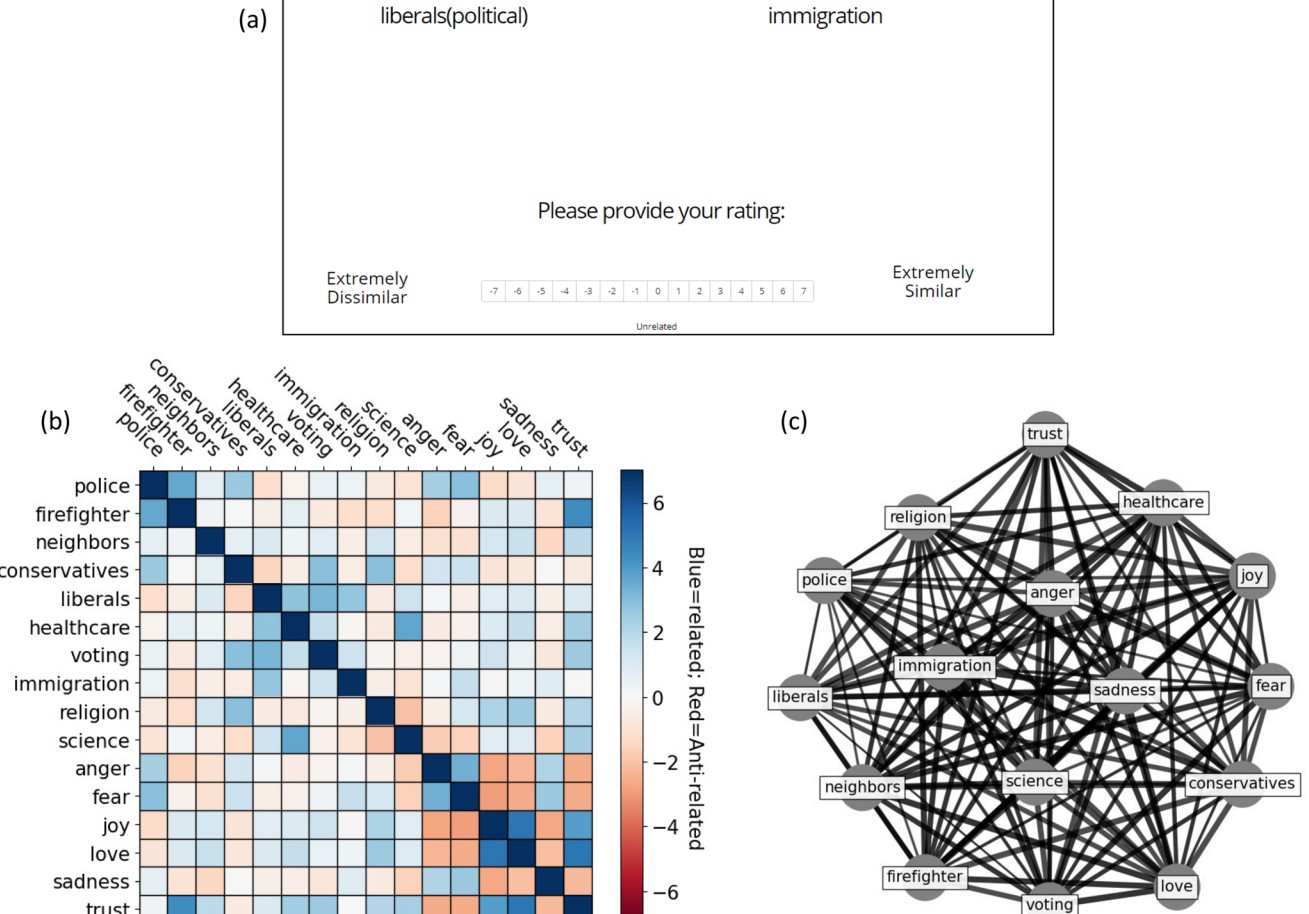

**Fig. 1 | Concept graphs generated using pairwise ratings of associability.** **a** Example of a single trial for obtaining pairwise concept rating. **b** Average association rating across the full sample for all 120 possible pairs. **c** Fully connected concept graph of inter-concept associations. This graph is based on the average similarity data across the full sample of analyzed data (N = 446). Each concept is represented as a node, concept-pair edge weights are used to tune distances between nodes.

## Experimental procedure

Concept pair associativity was measured using the Pairwise Rating Methods (PRaM)[22]. Participant-provided ratings, ranging from −7 to +7, were used to assess the (dis)similarity between pairs of concepts (Fig. 1a). A rating of −7 indicated a strong opposition between concepts, while +7 signified a high degree of relatedness. A rating of 0 suggested no relation between the concept pair. These ratings were gathered for every one of the 120 different combinations of the following 16 concepts: *police, firefighter, your neighbors, political conservatives, political liberals, healthcare, voting, immigration, religion, science, anger, fear, joy, love, sadness,* and *trust*. These socio-environmental concepts were selected based on previous research on attitude that showed discrepancies between Black and white Americans. *Firefighters* was an exception in that it was selected because there were no expected identity-based differences. The presentation order of the concept pairs was randomized, however, their left-right positioning was not. The mean probability of left positioning across the 16 concepts was 50% with a range of 0–100%. Although we do not suspect the left-right positioning of the concepts to influence ratings, future implementations of this task will aim to balance this.

After completing the PRaM, participants were asked to report their 3 most frequently consumed news sources. Participants who provided news consumption responses which did not provide sufficiently detailed information were excluded from the analyses; this was primarily due to participants listing exclusively algorithm-driven news sources (e.g., YouTube, Facebook, Reddit, X/Twitter). The political bias associated with each news source was assessed along a 5-point scale (Left (1), Left-leaning (2), Center (3), Right-leaning (4), or Right (5)) using All-Sides media bias metrics[27]. News consumption political bias was then computed by taking the mean across bias rating across the three listed news sources.

## Data analysis

To measure the internal reliability of the Pairwise Rating Method (PRaM) a split-half reliability measure was computed[22]. The reliability of the whole sample was 0.94; 0.94 for the subset of Black identifying participants and 0.92 for the subset of white identifying participants.

Identity and news source effects were estimated for each concept-pair using an edge-wise general linear model. Although a single multivariate model containing all 120 concept pairs could have been computed, due to statistical power constraints and our interest in individual concept pairs, regression modeling (with family-wise error correction) was computed on individual concept pairs. Model predictors included: racial identity (dummy coded such that Black / African American = 1); gender (dummy coded such that Female = 1); income (ordinalized by income bracket (see Supplementary Fig. 1 for bracket breakdown)); age; mean news consumption bias (see Supplementary Table 1 for full model results). Observed effects were compared against a null distribution generated for each concept-pair using a 10,000-iteration permutation test, where features were randomly scrambled on each permutation. Multiple comparisons were then corrected using a false-discovery rate (FDR) of 0.05. Age, gender (male or female), and income did not show significant differences for any of the 120 edges with exception of *voting-joy* for Age, r(444) = −0.16, p < 0.001, 95%CI [−0.25, −0.07], with older individuals rating the pair as more similar. Nonetheless, they were all included as nuisance terms in the model.

The relationship between concept uncertainty and news-bias effects for each concept pair were measured using Shannon's entropy. Entropy was calculated across participants for each concept-pair as $H = -\sum_{i=1}^{n} p(x_i) \log_2 p(x_i)$, where $p(x_i)$ is the probability that rating $i$ is selected for a specific concept pair. An offset sigmoid function was fit to the relationship between news-bias effects and entropy: $1/1 + e^{-a*(x-b)} + c$.

A subsequent mediation analysis[28] was computed to measure whether the effect of racial identity on edge-length was moderated by the bias in the news that people consume. Dummy coded racial identity ($x$) was used to predict concept-pair distances ($y$) treating bias in news consumption as a mediator ($z$). Statistical significance of the mediation test was also measured using a 10,000-iteration permutation test. A Sobel test provided converging

evidence of mediating effects. All indirect and direct effects were FDR corrected for multiple comparisons.

## Reporting summary

Further information on research design is available in the Nature Portfolio Reporting Summary linked to this article.

## Results

### Understanding concept graphs from individual ratings

Pairwise concept associability served as the basis for generating graphs of concept pairs for individual participants. The *associability* of a concept pair can be intuited as its weighted connection, that is the magnitude of semantic relatedness[29]. Concept-association ratings were obtained by asking participants to provide ratings (−7 to +7) of the (dis)similarity for a pair of concepts (Fig. 1a). Ratings of −7 represented anti-relatedness (i.e., opposites), ratings of +7 represented extreme relatedness, and 0 indicated that the pair of concepts were unrelated. Ratings were obtained for all 120 possible combinations of the following 16 concepts: *police, firefighter, neighbors(yours), conservatives(political), liberals(political), healthcare, voting, immigration, religion, science, anger, fear, joy, love, sadness, trust*. These 120 ratings constitute the complete set of associations for individual participants for this set of concepts (Fig. 1b). Concept-pair ratings were then converted to edge weights that were used to compute the length of unit vectors within a graph (Eq. 1).

$$e_{(i,j)} = 1 - \frac{(s_{(i,j)} - r_{\min})}{(r_{\max} - r_{\min})} \quad (1)$$

Where $s_{(i,j)}$ is the similarity rating for concept-pair$_{(i,j)}$ and $r_{\min}$ and $r_{\max}$ refer to the minimum (−7) and maximum (+7) values of associability (Fig. 1a). The resulting concept-pair edge weight $e_{(i,j)}$ ranging from (0,1) is proportional to associability, with values closer to 0 indicating greater similar associability and values closer to 1 indicating greater dissimilarity (Fig. 1c). Group variables (e.g., racial identity, partisan news consumption) were used to predict differences in individual concept edge-lengths. The subset of differing edge-lengths can then be represented as subgraphs (Fig. 2a). The configuration of the resulting subgraphs depicts a cognitive map for how individual participants or group of participants represent the relationships among this subset of socioenvironmental concepts.

### Differences in concept geometry across racial identities

Black and white identifying Americans differed in their associations of socioenvironmental concepts along several concept pairs (represented as nodes and edges comprising a subgraph; Fig. 2a). A concept node's centrality in this context indicates how frequently it differs between groups, based on the number of unique edges it is connected to. Therefore, a concept's centrality value reflects the proportion of pairs that feature the concept that show a statistical group difference. The socioenvironmental concept nodes that tended to serve as central hubs for the race-related edge-length differences including *religion* (centrality = 0.64), *conservatives* (centrality = 0.57), *science* (centrality = 0.50), and *police* (centrality = 0.43). In some cases, edge-lengths were shorter (i.e., indicating greater similarity) for Black participants than for white participants (e.g., *police-fear*; represented as blue edges in Fig. 2a) while in other cases edge-lengths were shorter for white participants than for Black participants (e.g., *voting-trust*; represented as red edges in Fig. 2a). For some of these concept-pairs, the difference in similarity ratings (Fig. 2b) reflects varying magnitudes of associability in the same direction (e.g., *police-fear*) while other similarity ratings reflected associability in antipodal directions (e.g., *police-trust*). Separating subgraphs for each racial group along edges with the strongest group effects reveals overall qualitatively similar topologies between the two groups (Fig. 2c, d). Although differences exist between racial groups, this analysis does not reveal to what extent do informational ecosystems may mediate these differences (contextualized as politically Left or Right partisan bias in news consumed).

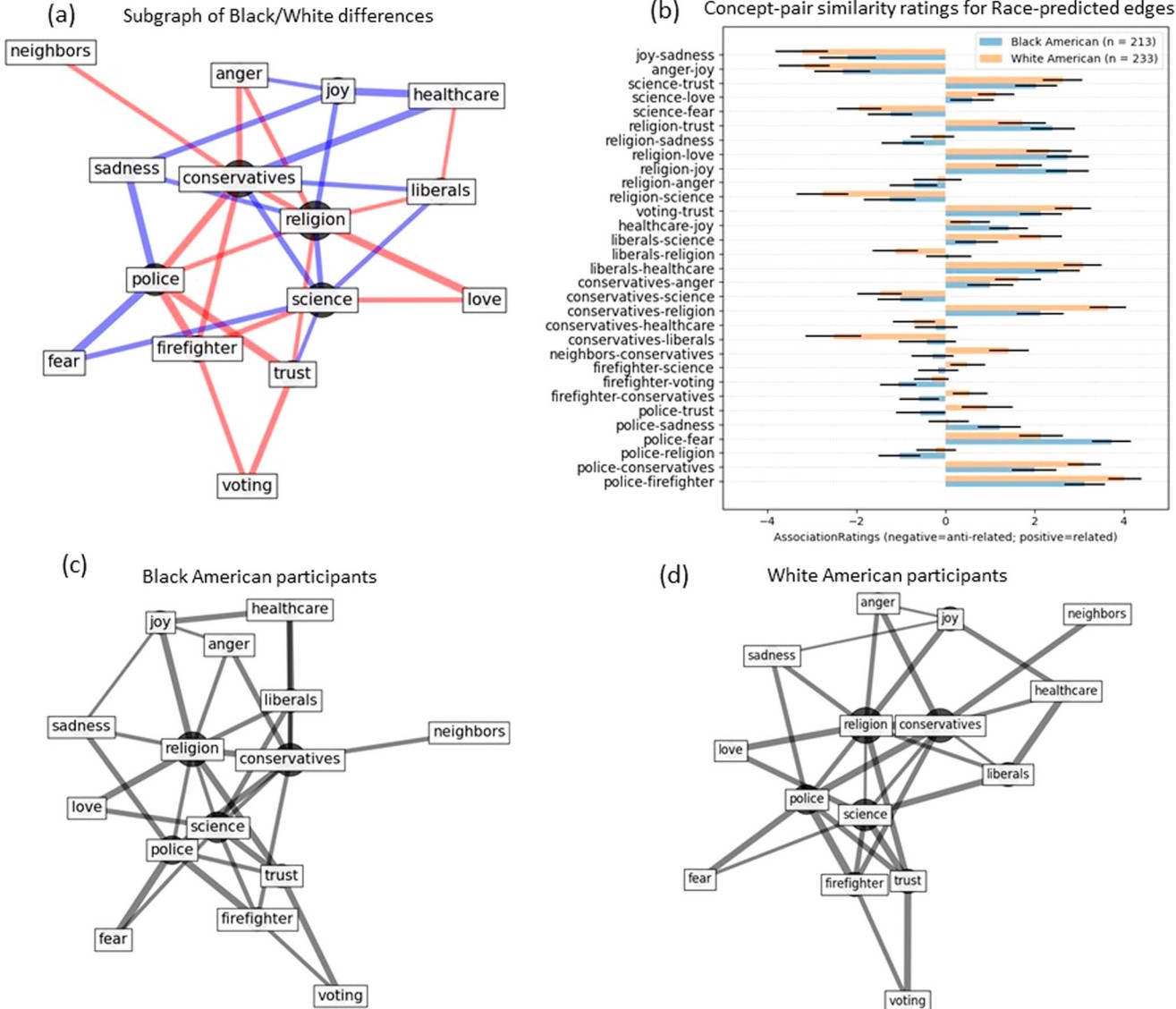

**Fig. 2 | Subgraph and group comparisons for race-differentiated concept associations. a** Subgraph of differing edges between Black and White participants. edges are for pooled data between Black and White participants ($N = 446$). Concept edges are represented by the edge width/length (wider/closer = stronger association; thinner/farther = weaker association). Edges with stronger association for Black participants are shown in blue and edges with stronger association for White participants are shown in red. **b** Bar plot showing association ratings for race-differing edges averages for Black and White participants separately. Error bars mark 95% CI boundary. **c, d** Subgraphs of race-differing edges for Black and White participants, respectively, for the same edges and nodes as panel a. Similarity data are pooled within each group.

## News bias is associated with greater uncertainty

One way news source could bias the association between concepts is by influencing the most pliable, i.e., uncertain, associations. We looked for a signature of this effect by measuring the Shannon entropy of concept pairs and seeing how this measure associated with news bias effects in concept relations. Regression coefficients of the effect of news bias on concept-pair associations were asymptotically related to an increase in entropy for concept pair association (Fig. 3a). The direction of the news bias effect (positive or negative regression coefficients) did not differ in their respective association with concept pair entropy (Fig. 3b). The concept pairs that were unaffected by news bias also had the lowest entropy, including pairs like *joy-love* and *love-trust*. Examples of concept pairs that showed high effects of news bias and high entropy include *liberals-trust* and *conservatives-trust*. Overall, there was a strong correlation between the news bias effects on a concept pair and that pair's across-subject entropy, with entropy explaining over a quarter of the variance in the news bias effect (fit: $a = 36.329$, 95% CI [6.86, 65.80]; $b = -0.004$, 95% CI [−0.03, 0.02]; $c = 2.729$, 95% CI [2.45, 3.01]; $R^2 = 0.271$).

Thus, greater uncertainty of an individual concept pair relationship meant it was more likely to also be associated with the political bias of an individual's news source. We next explored how this effect of information ecosystem could mediate our observed group differences in concept relations.

## News bias mediates identity-based associations

Finally, we are ready to move on to our primary hypothesis on how partisan bias in news consumption may mediate racial identity-based differences in concept geometry. Left- and Right- leaning news consumers had differing edge-lengths along 44 of the 120 possible concept pair associations (see Supplementary Fig. 2 for bar plot of association ratings between Left- and Right- leaning news consumers). Among the 44 concept pairs that differed by news consumption, 18 also differed between Black and white Americans in a seemingly systematic way. Specifically, for this common subset of 18 group-differing edges, Left-leaning news consumers and Black Americans showed similar patterns of concept pair associations, whereas Right-leaning news consumers and white Americans displayed similar patterns. Although

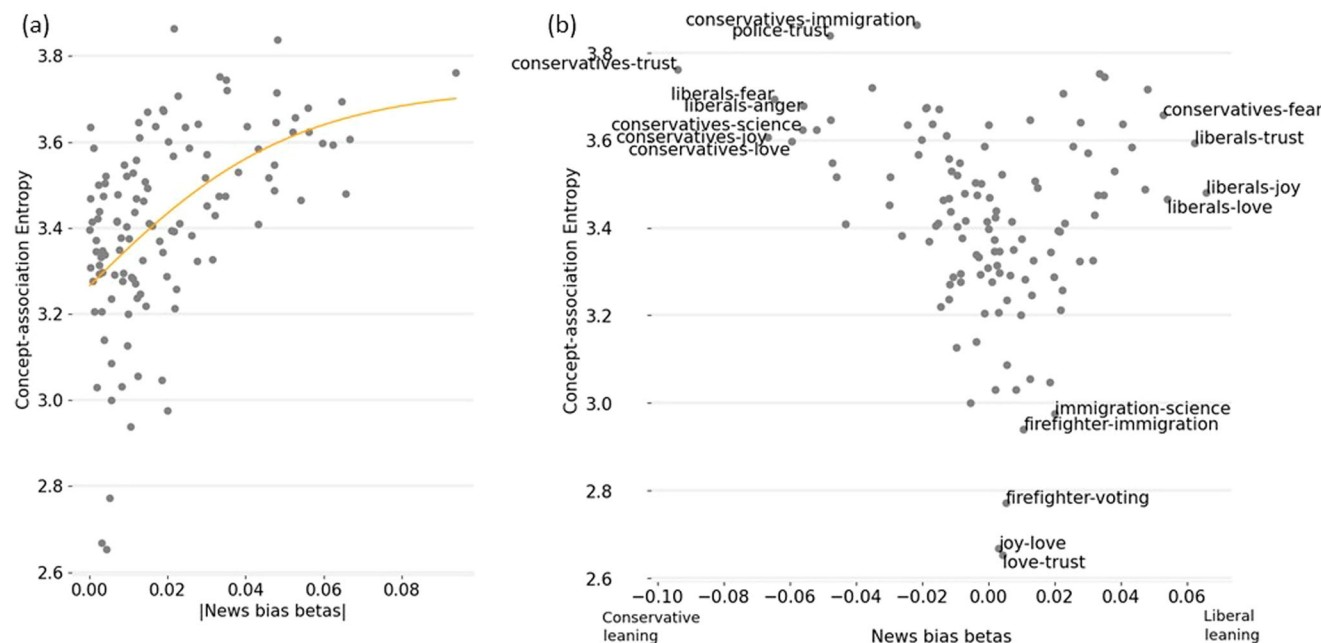

**Fig. 3 | Relationship between news bias effects and concept association uncertainty. a** The absolute value of news bias effects plotted against concept association entropy. **b** News bias regression coefficients plotted against concept-pair entropy. Concept pairs with extreme entropies or regression effects are labeled.

the total sample of participants tended to consume Left-leaning news (Fig. 4a; mean bias = 2.29, 95%CI [2.18, 2.41]), there were significant differences in the direction of partisan news consumption between Black (mean bias = 2.04, 95%CI [1.89, 2.19]) and white participants (mean bias = 2.52, 95%CI [2.36, 2.69]; Fig. 4b), t(444) = −4.216, p < 0.0001, d = −0.4, 95% CI [−6.18, −2.25]. A significant correlation between race and news bias consumption, r = −0.195, p < 0.0001, 95%CI [−0.28, −0.1], suggests that Black identifying participants consume more Left-leaning news and white identifying participants consume more Right-leaning news (although it should be noted that, in the aggregate, the magnitude of this difference is only about 0.5 points on the news bias rating scale). A follow up statistical mediation analysis revealed that news bias partially mediated 18 of the original 31 race-differing edge-lengths, while 3 of the remaining 13 edge-lengths were completely mediated by news bias (for a schematic of the mediation model and full model results see Supplementary Fig. 3 and Supplementary Table 2, respectively). These fully news-mediated concept-pairs included: *conservatives-healthcare*; *religion-anger*; and *religion-love*. Figure 4c shows a subgraph of the partially mediated edges subset from the subgraph of race-differing edges (Fig. 2a). Figure 4d shows the subgraph of the race-differing edges, excluding the 3 edges completely mediated by news effects. Concept relations differed more across Left and Right news consumers than across Black and white racial groups, as evident by the number of concept pairs that show differences (i.e., 44 vs 31). Moreover, while news consumption partly explains racial differences in the overlapping 18 concept pairs, these differences persist even after accounting for news consumption.

## Discussion

We found that identity and informational ecosystems play a critical role in how we think about our societal environment. Specifically, Black and white identifying participants had consistent inter-group differences in their concept graphs for socioenvironmental concepts. A subset of these race-differing concept-pair edges were partially mediated by the partisan bias in the news participants consumed. That is, news consumption partly explained racial differences in several overlapping concept pairs; however, these differences persist even after accounting for news consumption. However, for all but three edges, the race-differing concept-pairs that were statistically mediated by news bias continued to show differences across groups, suggesting that news bias does not eliminate associative differences

across race. Independent of race, over a third of the edges in concept graphs (44 of 120) were associated with partisan biases in news sources and this effect correlated with the overall entropy of the concept-pair, suggesting that those associations with the greatest uncertainty were also the most susceptible to the bias of information sources. These findings suggest that there are associative differences in concept graphs across identity and informational ecosystems. Although people's values and opinions factor into their ratings of conceptual similarity, concept associations reflect the semantic structures that sculpt and filter an individual's view of their reality. In this way, a set of concept associations reflects a semantic knowledge that is inclusive of both what people know and how people feel.

What does it mean for white and Black Americans to differ in their concept maps for socioenvironmental concepts? It is first important to note that this study is insufficient in establishing a causal relationship between any facet of identity and concept association. Even if our observed effect between news source and group biases in concept relations were causal, it only explains a portion of the variance in differences between Black and white Americans. It is also important to note that the construct of racial identity is not innate, nor can any racial identity be reduced to an essential set of views or characteristics[30]. One possible explanation for racial differences in the representation of socioenvironmental concepts may be the historical and empirical disparities in how Black and white Americans have experienced public-serving institutions. Although there have been positive trends in equalitarian attitudes in the United States since the 1940s[31,32], people of color are at greater risk of being killed by police use of force[33,34], receive more severe treatment plans by clinicians[35], and have a greater likelihood of being targeted for voter disenfranchisement[36]. These empirical studies suggest that racial and ethnic minorities have largely different life experiences in America relative to white Americans, which would be expected to manifest on how they think about associated concepts. This could be an explanation for a large part of the variance of our group effect that is not explained by biases in news consumption.

Consistent with previous research, we also found that political leaning diverged between Black and white racial identity lines, from Left to Right respectively (Fig. 4b)[17]. One possible explanation for this divergence is that people tend to consume media that aligns and reinforces existing values, goals, and ideas[37]. Evidence to support this claim at a group level could be that despite a difference in Left- versus Right- leaning partisan news

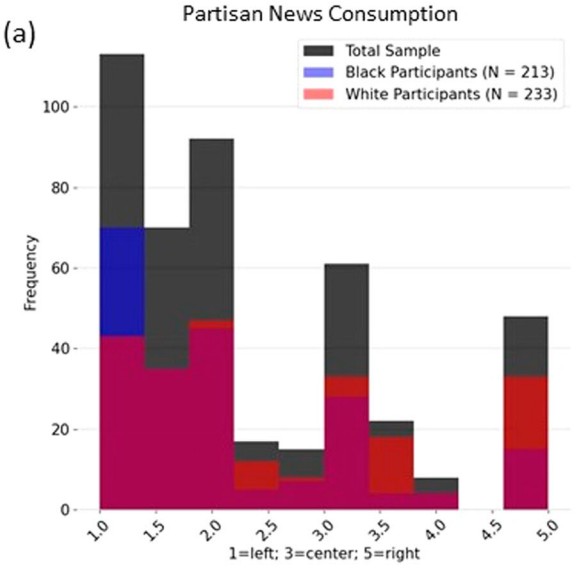

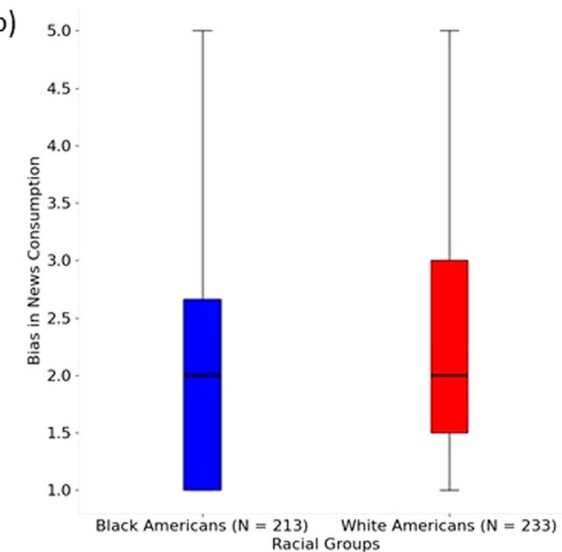

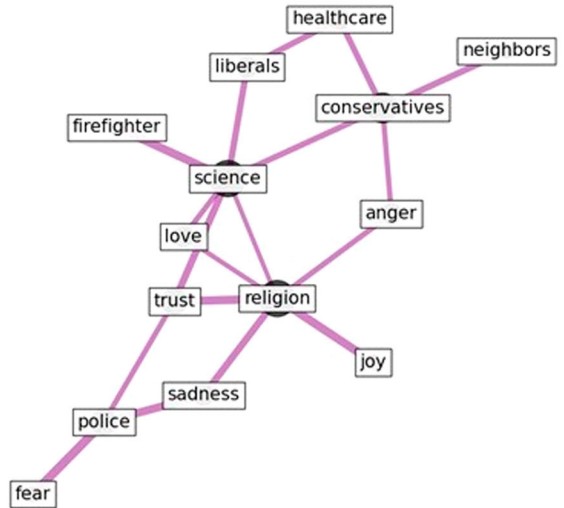

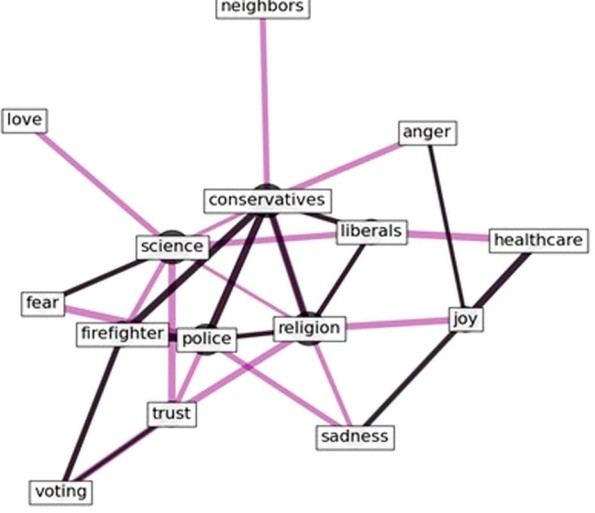

**Fig. 4 | Partisan news consumption and its mediation of race-differing concept associations.** Partisan news consumption partially mediated several edges that differed between Black and White participants. **a**, **b** Although the total sample generally consumed Left-leaning news, White participants consumed more Right-leaning news than Black participants. **c** These news-mediated edges can be represented as a subgraph of the race-differing effects. Only 3 of the original 31 edges that differed between Black and White participants were completely mediated by news consumption. These edges are: *conservatives-healthcare*, *religion-anger*, and *religion-*

*love*. **d** Subgraph of all race-differing edges excluding the 3 edges that were completely mediated by news consumption. Purple lines denote edges that are partially mediated by news consumption but also have a direct effect of race, that is, they are partially mediated by news consumption. Dark grey lines denote race differing edges that did not differ by news consumption. The nodes that served as hubs include: *religion* (centrality = 0.5), *conservatives* (centrality = 0.5), *science* (centrality = 0.5), and *police* (centrality = 0.43).

---

consumption across racial groups, news bias only completely mediated 3 of the 31 concept pairs and partially mediated 18 of the 31 concept pairs across racial identity and concept association. One example of a partially mediated concept pair was *police-trust*. That is, even though news consumption explained part of the differences in association between racial identity and *police-trust*, racial identity-based differences were maintained even when controlling for news effects. These results suggest that news consumption likely does not invalidate identity-based experiences and associations between concepts. The concepts that were hubs for mediated news effects included *police*, *religion*, *science*, and *conservatives*. Media effect theories demonstrate that people are limited in their capacity to attune to information and often self-select media messaging that connects with, rather than challenges, their own experiences[38–40]. In this way, participants may be self-selecting the news that they consume such that it is consistent with personal experiences or attitudes towards hub concepts that amplify

differences across racial identities. Further work is needed to elucidate the causal direction(s) in this effect.

The effect of partisan bias of the news sources in our study was both a stronger influence on concept geometries than racial identity and partially mediated the racial group effects in concept geometries. This associative difference across news consumption, regardless of specific partisan bias, brings into focus the relative impact of informational ecosystems on an individual's understanding of the world. In 2022, the average American spent 7 h and 19 min engaging with digital media per day[41], with 33% of Americans preferring to get their news from televised programming and 53% of Americans preferring their news from digital devices such as smartphones, computers, or tablets[42]. Of this 53% of Americans who prefer digital devices, 43% utilize news websites. News sources are thought to frame concepts shared by a group of individuals and specify criteria for subsequent evaluation[8], which is largely supported by our observations here. However,

as with any group identity, political ideological views are not monolithic. Measures of news consumption do not describe the specific quality of an individual's views such as beliefs on limited government and cultural conservativism[43].

It is worth noting that the concept pairs most susceptible to partisan news bias were also the most uncertain, as measured by Shannon entropy. This effect was largely pronounced in how participants think about liberal and conservative political ideologies (e.g., *conservatives-trust* and *liberals-trust*). This political ideological divisiveness is further supported by an increase in representational distances across ideological lines for the subset of concept-pairs that included a concept referring to a political group (i.e., *conservatives* or *liberals*; see Supplementary Fig. 4 for scatterplot of news effects and inter-partisan representational distances). One possible mechanism for explaining the relationship between across-participant entropy and news effects could be a limitation in information processing. Limitations in information processing capacity have been theorized to bottleneck viewers to attune to selective information[44,45]. Although past media effect research has largely focused on televised programming, digital media consumption has continued to increase yearly from 214 min daily in 2011 to 438 min daily in 2022[41]. This increase in digital media consumption speaks to the growing accessibility and quantity of available information despite limited capacity for information processing in a world of readily present and accessible information. In the context of this study, one possibility for the increased uncertainty in concept pairs containing socio-environmental concepts is a consequence of contradictory conceptual associative messaging from partisan sources. Although our findings are consistent with past research suggesting that news bias effects influence conceptual representations[8], it is possible that the relationship between news bias effects and concept pair uncertainty could be mediated by unknown factors.

Concepts serve as the units of meaning that organize our understanding of both the internal (mental) and external (physical) world. How concepts are organized together sets the framework for making decisions on how to act in the world. Critically, both concepts and their relative associations are not immutable. Just as social biases are contingent on context[5] so are concepts. Any identity (racial, political, or otherwise) cannot be reduced to a prescribed set of associations, rather, identity can be thought of as reflecting social and societal contexts (including experiential or informational[46]; which alter the probability of the formation of certain associations. Here we show that concept associations differ along racial identity and partisan news consumption boundaries. Notably, the magnitude of the effect of news consumptions is considerably greater than that observed by racial identity. The relative strength of the news consumption effect over racial identity is highlighted both by the number of concept pairs with differing edge lengths (44 compared to 31) and by the magnitude of rating differences observed along those differing concept pairs (Fig. 2b; Supplementary Fig. 2). This difference in impact suggests that informational ecosystems play a larger role than group membership for semantic cognition. Although this study does not specify mechanisms for differences between white and Black racial identities and between partisan news consumers, it provides an empirical basis for subsequent investigation into possible mechanisms driving differences in concept geometries. Specifically, how do our experiences or the experiences of groups we identify with influence our perceptions of the society we are nested in and how are the resulting associations causally influenced by the informational ecosystems we are exposed to?

## Limitations

Although participants recruited through crowdsourcing platforms are more representative of the broader U.S. population than locally recruited college samples, they remain often less diverse and suffer from concerns about response quality[47]. Crowdsourced participants tend to be more educated, younger, and predominantly white identifying relative to the general population[48]. Moreover, crowdsourced participants tend to generally be politically left leaning, as we observed in the distribution of news bias scores in our sample. Yet, the values and personality traits typically held by conservative and liberal individuals recruited through crowdsourcing tend to reflect the general population[49]. To mitigate issues of representativeness, selective recruitment was implemented to target groups of interest to allow for large samples of participants to be recruited within each group. Moreover, participants were recruited through Cloud Research Connect or through Cloud Research's TurkPrime Toolkit which tend to have higher response quality relative to recruitment through MTurk directly[50]. In summary, although steps were taken to mitigate unrepresentativeness within the sample, it is possible the results presented here deviate from the general population.

## Conclusions

Racial identity and informational ecosystems—defined through news consumption biases—play critical roles in shaping the semantic associations between socioenvironmental concepts. Black and white identifying Americans exhibited reliable group differences in their conceptual relationships, potentially reflecting broader disparities in life experiences and information exposure. Partisan news consumption partially mediated these differences, with politically polarized sources exerting a stronger influence on concept geometry than racial identity alone. Notably, concept pairs that reflected higher levels of uncertainty were more susceptible to the effects of partisan bias, highlighting the plasticity of concept associations under informational influence.

Despite news bias accounting for some inter-group variability, racial differences in concept associations persisted, suggesting that lived experiences remain critical to cognitive representations. These findings demonstrate the interplay between identity, media consumption, and cognitive structures, highlighting the need for further research into the mechanisms by which lived experiences and media messaging interact to influence social cognition within a larger ecological context.

## Data availability

All de-identified data are openly available via corresponding authors' lab GitHub: https://github.com/ExCaLBBR/ExCaLBBR_Projects/tree/main/SocioenvironmentalGeometry.

## Code availability

All code for data processing, analyses, and visualization were developed in Python and are openly available in the form of Jupyter notebooks via corresponding authors' lab GitHub: https://github.com/ExCaLBBR/ExCaLBBR_Projects/tree/main/SocioenvironmentalGeometry. All code is executable via Google Colab such that no local software or dependencies are required.

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

## Acknowledgements
The authors would like to thank Jayln Wiliams, Dr. Kevin Jarbo, Dr. Cassie Eng, Dr. Sylvia Perry, and the CoAx lab for guidance and mentorship. This project was funded through the Presidential Postdoctoral Fellowship program at Carnegie Mellon University and by AFOSR/AFRL award FA9550-18-1-0251. The funders had no role in study design, data collection and analysis, decision to publish or preparation of the manuscript.

## Author contributions
R.V. contributed to the study conceptualization, study design, data collection, data management, analysis conceptualization, analysis implementation, result visualization, and writing. T.V. contributed to the study design, analysis conceptualization, writing, supervision, and funding acquisition.

## Competing interests
The authors declare no competing interests.
