## [Peer Review file · Communications Psychology]

Informational Ecosystems Partially Explain but do not Eliminate Differences in Socioenvironmental Conceptual Associations between U.S. American Racial Groups

Corresponding Author: Dr Roberto Vargas

Version 0:

Decision Letter:

Dear Dr Vargas,

Thank you for your patience during the peer-review process. Your manuscript titled "Identity and Informational Differences in Concept Relations for Social-Societal Institutions" has now been seen by 2 reviewers, and I include their comments at the end of this message. They find your work of interest but raised some important points. We are interested in the possibility of publishing your study in Communications Psychology, but would like to consider your responses to these concerns and assess a revised manuscript before we make a final decision on publication.

We therefore invite you to revise and resubmit your manuscript, along with a point-by-point response to the reviewers. Please highlight all changes in the manuscript text file.

Editorially, we consider the criticism regarding the representativeness of the sample one that needs to be addressed. While we welcome studies conducted with MTurk (or similar platforms), the demographics of the sample should be described comprehensively and evaluated in terms of its representativeness. Likely differences that cannot be accounted or controlled for should be discussed as a limitation. In general, the reviewers' comments indicate that the work needs to be improved in its presentation of the background literature as well as in terms of the clarity with which the present study and its methodology are described. As you address the novelty concern, please note that we discourage novelty claims (describing the study as novel, first, etc). Instead, we ask that you explain how your work confirms, contradicts, or extends the existing body of relevant literature.

I am attaching an Editorial Requests Table that details critical reporting requirements for the revised manuscript. Please attend to each item and ensure your manuscript is fully compliant. We are requesting that your manuscript aligns with these requirements as this facilitates the evaluation of your manuscript, reducing delays in re-review and potential future acceptance. If your revised manuscript is not aligned with these requests on major issues, such as those concerning statistics, it may be returned to you for further revisions without re-review. Additional information can be found in our style and formatting guide <https://www.nature.com/documents/commspsychol-style-formatting-guide-accept.pdf>>Communications Psychology formatting guide.

Please use the following link to submit your

- revised manuscript,
- point-by-point response to the referees' comments,
- cover letter (as a separate document),
- the Editorial Policy Checklist (see below),
- the Reporting Summary (see below), and
- the completed Editorial Request Table (attached):

Link Redacted

Best regards,

Yafeng Pan

Yafeng Pan, PhD
Editorial Board Member
Communications Psychology
orcid.org/0000-0002-5633-8313

REVIEWER EXPERTISE:

Reviewer #1: Computational social science (Pair-wise Rating Methods)

Reviewer #2: Group identity

REVIEWER REPORTS:

Reviewer #1 (Remarks to the Author):

This paper had participants who either identify as Black or White Americans rate the similarity of 16 concepts and identified the concept pairs on which the two groups systematically differ. Several of these racial group differences are then found to be partially mediated by the bias of the news sources the individual participants consume. The paper suggests that there exist systematic semantic differences between existing social groups, which are partly due to the groups' different (media) experiences. These results extend previous findings which have established similar differences between age, gender, and education groups. I suspect the results will be of interest to many readers both in social and cognitive psychology, as well as in sociology and media studies. The paper communicates an important message about the impact of media biases.

The data collection is sound, the figures provide face validity to the results, and the conducted analyses are sophisticated. There also lies the difficulty with the work. Its presentation in a rather short format makes it difficult to fully interpret and obscures some elements of the work. My main recommendations therefore reside at the level of the presentation, but I also provide several smaller comments. I do applaud the authors for making the data and code publicly available. It offsets some of the sparseness in the presentation.

MAJOR COMMENTS:

Most people studying semantic cognition will not be surprised to learn that experiences shape semantic relationships. To appeal more to this audience, I recommend the authors take a little more space to explain the novelty of their contribution, which in my opinion lies in the demonstration of an effect of media consumption on semantic cognition and the demonstration of semantic differences between racial groups, which hasn't been shown as far as I know. To appeal to social psychologists more, the authors could make a stronger case that the relationships that are being studied are semantic ones and not opinions or values. I don't think social psychologists would be surprised to learn that the opinions of different racial groups differ on matters like voting, immigration or science. I don't think that the fact that you ask about the similarity of concepts would convince everyone that you are tapping into semantic knowledge if the latter is understood as "factual" knowledge. Related to this, the authors often use the term 'associability' to refer to the investigated relationships. In the semantic cognition literature, this refers to a rather specific type of semantic relation. I am not sure the authors are aware of this and want to argue that it is this specific type of semantic relation that is being studied/impacted?

In several places throughout the paper, the authors speak about the geometry of inter-concepts associations or concept graphs. If I understand the analyses correctly, the authors do not build geometrical or graph representations of the similarity data they obtained and compare these between groups, but rather conduct statistical inferences on transformed similarity data for individual concept pairs and use graphs (network representations) to illustrate the resulting differences simultaneously for the different pairs. Because of this, the emphasis on graphs and geometrical representations could perhaps be toned down a bit, because there is an extensive literature in which such representations are built and investigated, but that does not seem to be done here. It also raises the question why the authors decided to conduct their

analyses at the level of concept pairs rather than at the level of all inter-concept relationships simultaneously? Perhaps they did not want to make any representational assumptions? In any case, I would appreciate an indication of the relative strengths and limitations of both approaches. Somewhat related: I hadn't come across the use of subgraphs to highlight edge differences between groups before. Is this a new visualization technique the authors came up with or has this been tried before? I ask, because it find it promising, but also wonder how good a job it does conveying the differences. For instance, to appreciate the differences between the groups in Figure 2 we need to integrate across a single graph representation for the aggregate data (panel a) and two separate graph representations (one for each racial group in panels c and d).

I appreciate that the authors' main interest lies with racial group differences and how these are mediated by news consumption. I did find it interesting to see that news consumption appeared to yield more extensive effects than race, at least if one looks at the number of concept pairs that differed in the two comparisons (44 pairs vs 31 pairs). Do the authors consider this a substantial finding? For instance, can it be taken to indicate that experience is more important than group membership for semantic cognition? Intuitively, that would make sense to me.

MINOR COMMENTS:

- p. 3: It was not immediate clear to me what you mean when you write that schemas should follow specific criteria for subsequent evaluation.
- p. 3: The observation that selective consumption of media may enhance beliefs in conspiracy stories is a rather extreme example of the effects of media consumption on cognition. Can you provide a more moderate example as well?
- p. 4: Perhaps explicitly indicate that the graph in panel c is based on the average similarity data across the full sample.
- p. 5: The notion of centrality is introduced in passing here. I recommend explaining further how it is operationalized. I take it the discussion of centrality is not meant to convey that the hubs differ between groups, but to signal which hubs are central in the relationships that differ between the groups? What can we learn from the observation of these specific hubs?
- p. 6: The scale values along the horizontal axis of panel b in Figure 2 are missing.
- p. 7: Perhaps it would be worth while spelling out explicitly that the relationship between entropy and news bias betas is not a necessary one. Obviously, for any predictor to have a significant effect there needs to be variability (entropy) in the variable that is being predicted, but not every predictor will be able to explain this variability. (Maybe this is obvious to most readers of the journal, but I had to think about this for a moment to ascertain myself that no spurious relationship was being reported.)
- p. 7: I found the passage that starts with "Of those 44..." and ends with "...White Americans, respectively." difficult to follow, while it is very important for the paper. Could you please rephrase, elaborate, or provide an example in support?
- p. 8: Which test was conducted to establish the bias difference in news consumption between Black and White participants? Note that often in the paper the word 'across' is used where I would use 'between' (as in the example of the bias test). To my non-native ear at least, the use of 'between' conveys better what is done.
- p. 8: Would it be possible to report the outcome of the mediation analyses somewhere, for instance in the supplementary info? This would make it easier to appreciate the difference between unmediated, partially mediated, and fully mediated race differences.
- p. 9: Figure 4d appears to use a form of color coding but this isn't explained. For consistency, the authors could also consider using the same representation format as in Figure 2.
- p. 10: The concepts police, religion, and science are here identified as hubs for the mediated news effects, but I don't think this has been established yet in the results section (other than shown in Figure 4).
- p. 11: Please elaborate on the significance of the following sentence: Measures of news consumption do not describe views on dimensions used to characterize political conflict including limited government and cultural conservatism.
- p. 12: What was the rationale behind the choice for the 16 concepts? The title hints at the relevance of social-societal institutions, but this isn't further explained. A word on presentation order of concept pairs and words within pairs during the pairwise rating task would be appreciated as well. I applaud the authors for computing the reliability of the similarity ratings. This is a pre-requisite for its visual representation.
- p. 12: I would appreciate it if the outcome of the general linear model would be reported somewhere, for instance in the supplementary info. The model includes mean news consumption bias, but I don't think we learn anything about this variable's contribution in the model.
- p. 13: Is the formula for Shannon's entropy traditionally not preceded by a minus sign?

I always sign my reviews.

Steven Verheyen (verheyen@essb.eur.nl)
Erasmus University Rotterdam

Reviewer #2 (Remarks to the Author):

This paper uses an interesting and novel (at least to me) approach to explore how associations between concepts vary across groups. The argument—and the evidence—suggests that groups of people may develop different “concept maps” of the world. Different groups not only evaluate different individual people, ideas, and things differently, but also perceive different relations among those objects. These group differences seem to be partly attributable to (or at least associated with) group members' tendency to reside in different “information ecosystems.” Such as, in this case, relatively liberal versus conservative media environments in the U.S.

I think this question of how different groups perceive and map reality differently is very interesting. It reminds me of research on political ideology, which has long grappled with similar questions of “what goes with what” and for whom, but the authors present a method that could be adapted to explore relations between an infinite variety of idea elements that may or may not be political. I think their data would be interesting to many readers.

I do have some concerns and suggestions:

1) Because the work is at least partly descriptive and not experimental, the sample coming from MTurk is a potentially significant limitation. I would like the paper to include more discussion about how representative this sample is of the intended target population(s). This is especially important given the authors’ interest in subgroups; the question is not just how MTurkers differ from the U.S. population as a whole, but how White and Black MTurkers differ from White and Black Americans, and whether and how their concept maps would likely differ as a result. I defer to the editor on the importance of representativeness for publication, but I do think this limitation merits more page space, with a clearer presentation of what is safe vs not safe to conclude from this sample.

2) The authors mention that they analyzed “various facets of identity (including race, gender, income, and age).” I realize space here is limited, but even if the authors don’t have time to cover what all of the group-related differences were, I would be curious to know *whether* concept maps differed depending on age, gender, or other group identities.

3) The discussion talks about this some already (p. 10), but I would like to see a bit more attention (in the results or intro) to what “partial mediation” (or even just controlling for media environment) means in this context, with more done to unpack it in the quantitative analyses. I think this is a really vital question for this paper – Among conservative news consumers (or among liberal news consumers), are Black and White respondents very similar in their concept maps? Or still pretty different? Answering this question could help clarify the extent to which group differences in concept maps are the product of media ecosystems versus lived experience (or any other group differences not associated with the political slant of the news people watch). Attending to other group differences could also help answer this question. For example, did any other groups differ from one another in their political orientation as strongly as White and Black respondents did, and did they show similar patterns of mediation? Did other groups with identical political views end up with identical concept maps?

4) Some of the methods and results could be clarified for readers who (like me) have limited knowledge of the network analyses that the authors employ. Specifically:

- a. The authors provide a lengthy and helpful definition of “edge length,” and then provide figures with both edge length and edge width, which, to my untrained eye, appear to quantify basically the same thing (associations between constructs) in opposite directions. What is the difference between length and width?
- b. What is “Shannon entropy”?

5) Minor concerns:

- a. I’m not sure whether “bias” is the right word to use for the overall ideological slant of news media, given that neutrality may sometimes indicate bias too. Maybe this is just the term that AllSides gives its variable, though?
- b. On page 8, the authors write “the bias of news sources that an individual gets their information from does appear to influence expression of group differences in concept relationships.” This is probably too strong a causal claim.

EDITORIAL POLICIES

We ask that you ensure your manuscript complies with our editorial policies and reporting requirements.

To that end, we require revised manuscripts to be accompanied by two completed items: a reporting summary that collects information on study design and procedure, and an editorial policy checklist that verifies compliance with all required editorial policies.

- <https://www.nature.com/documents/nr-reporting-summary.zip>>Nature Research Reporting Summary
- <https://www.nature.com/documents/nr-editorial-policy-checklist.pdf>>Editorial Policy Checklist

All points on the policy checklist must be addressed. Your revised manuscript can only be sent back to the referees if these checklists are completed and uploaded with the revision.

Notes: If you have submitted a Stage 1 Registered Report, Review, Primer, Comment, or Perspective you do not need to submit these forms. If you have already submitted these forms, you may disregard this request.

* TRANSPARENT PEER REVIEW: Communications Psychology uses a transparent peer review system. This means that

we publish the editorial decision letters including Reviewers' comments to the authors and the author rebuttal letters online as a supplementary peer review file. However, on author request, confidential information and data can be removed from the published reviewer reports and rebuttal letters prior to publication. If your manuscript has been previously reviewed at another journal, those Reviewers' comments would not form part of the published peer review file.

Communications Psychology is committed to improving transparency in authorship. As part of our efforts in this direction, we are now requesting that all authors identified as 'corresponding author' create and link their Open Researcher and Contributor Identifier (ORCID) with their account on the Manuscript Tracking System prior to acceptance. ORCID helps the scientific community achieve unambiguous attribution of all scholarly contributions. You can create and link your ORCID from the home page of the Manuscript Tracking System by clicking on 'Modify my Springer Nature account' and following the instructions in the link below. Please also inform all co-authors that they can add their ORCIDs to their accounts and that they must do so prior to acceptance.
<https://www.springernature.com/gp/researchers/orcid/orcid-for-nature-research>

Version 1:

Decision Letter:

Dear Dr Vargas,

Thank you for your patience during the peer-review process. Your manuscript titled "Identity and Informational Differences in Concept Relations for Social-Societal Institutions" has now been seen by 2 reviewers, and I include their comments at the end of this message. We are interested in the possibility of publishing your study in Communications Psychology, but would like to consider your responses to these concerns and assess a final revised manuscript before we make a final decision on publication.

We therefore invite you to revise and resubmit your manuscript, along with a point-by-point response to the reviewers. Please highlight all changes in the manuscript text file.

Editorially, we ask you to further engage with the concerns regarding the representativeness of the sample as raised by Reviewer 2. Please address the concerns raised by Reviewer 2 regarding the interpretation of the mediation results and the request for the conducting and/or reporting additional analyses regarding the similarity of Black and White respondents in their concept maps holding news source constant. Please also address Reviewer 1's request for additional clarity regarding the methods and interpretation of the findings.

Please attend to all applicable requests in the attached editorial checklist. In particular, we draw your attention to our guidance on the manuscript format (e.g., Methods should precede Results) and our guidance on statistical reporting.

I am attaching an Editorial Requests Table that details critical reporting requirements for the revised manuscript. Please attend to each item and ensure your manuscript is fully compliant. We are requesting that your manuscript aligns with these requirements as this facilitates the evaluation of your manuscript, reducing delays in re-review and potential future acceptance. If your revised manuscript is not aligned with these requests on major issues, such as those concerning statistics, it may be returned to you for further revisions without re-review. Additional information can be found in our style and formatting guide <https://www.nature.com/documents/commspsychol-style-formatting-guide-accept.pdf> Communications Psychology formatting guide.

Please use the following link to submit your

- revised manuscript,
- point-by-point response to the referees' comments,
- cover letter (as a separate document),
- the Editorial Policy Checklist (see below),
- the Reporting Summary (see below), and
- the completed Editorial Request Table (attached):

Link Redacted

Best regards,

Yafeng Pan

Yafeng Pan, PhD
Editorial Board Member
Communications Psychology
orcid.org/0000-0002-5633-8313

REVIEWER EXPERTISE:

Reviewer #1: Computational social science (Pair-wise Rating Methods)

Reviewer #2: Group identity

REVIEWER REPORTS:

Reviewer #1 (Remarks to the Author):

I was Reviewer #1 of the original manuscript. I appreciate the authors' response to my comments and find the revised manuscript much clearer than the original. I have a few outstanding comments, but these can probably be addressed through minor revisions.

On p. 3, the introduction of semantic relationships feels rather abrupt. The preceding opening paragraph talks about ideas, values, and goals, which are typically the focus of social psychology studies in which racial groups have already been shown to have different attitudes toward social-societal institutions. The second paragraph does not explicitly indicate whether the authors think the semantic relationships they study should be considered different from these attitudes or not. In the discussion (p. 11) this becomes clearer, but I would recommend communicating this earlier because it signals the novelty of the contribution. At some points in the manuscript, the authors do not seem to make a distinction at all. On p. 10, for instance, they speak about their results in terms of "racial differences in attitudes on socioenvironmental concepts".

I still feel talking about concept graphs and concept maps is unnecessary and somewhat obscures the fact that the analyses are performed at the level of individual concept pairs instead of across the entire stimulus domain. I would find the text easier to digest if it were just to speak about inter-concept semantic associations. For instance, in the abstract one could eliminate the graph language without affecting the content: "Here we set out to see how inter-concept semantic associations differ between social groups, based on racial identity, and how this effect is mediated by information ecosystems, contextualized as news sources." The graphs could be identified in the method section as a way of visualizing the results.

MINOR COMMENTS:

- p. 5: Is there a straightforward way to interpret the values for centrality? Does the centrality value perhaps reflect the proportion of pairs that feature the concept which show a statistical group difference?

- p. 8: How are left- and right-leaning news consumers defined? Is it based on the center scale value or on a median split?

- p. 9: 'firefighter' was included because the authors expected no identity-based differences for this concept, yet it did show race differences. Personally, I am not surprised that it shows a group difference, but perhaps the authors want to say something about it, since its inclusion seems to be based on methodological considerations?

- p. 11: "The effects of partisan news consumption in our study were also proportional with the amount of uncertainty with which people consider socioenvironmental concepts." I believe 'pairs' should be added to this sentence, as the relationship pertains the concept pairs, not the individual concepts. Similar for the sentence later in the paragraph that reads "increased uncertainty in socioenvironmental concepts".

- p. 13: SIX should read SI5.

- SI1: "Associativity ratings are pooled across..." Here the authors could also add how they determined left- vs right-leaning news consumers (see earlier comment).

- SI4: Perhaps the authors could also provide an indication of the relationship between race and gender.

I always sign my reviews,
Steven Verheyen
Erasmus University Rotterdam

Reviewer #2 (Remarks to the Author):

I appreciate the substantial revisions that the authors have made to incorporate feedback from me and the rest of the review team. I think the manuscript is significantly improved. I have just a couple remaining suggestions.

First, I was glad to see the new "limitations" paragraph about the sample. That said, I think it would be helpful if the authors could talk more specifically, however briefly, about the racial subgroups that are the focus of their analyses. It is good to note how MTurkers tend to differ from Americans in general, but what I was particularly curious about here was how their Black respondents differed from Black Americans in general and how their White respondents differed from White Americans (e.g., in income, education, political attitudes). If this information is available to the authors, it would provide useful context and help inform readers as to the likely generalizability of the results. The sentence "our sample of Black and White participants show trends in political leaning consistent with past research (Gilens, 2023)" is an example of this type of information - this was helpful. I assume that by this the authors meant that the Black participants were generally more liberal than the White participants.

As the authors point out, this sample limitation is far from unusual. But given the goal of the project – to describe differences in how Black and White Americans perceive the world – I think it is unusually important in this study to make clear precisely whose perceptions are being described.

Second, I still think the authors could go a bit further to contextualize and explain their mediation results. This is a relatively minor quibble about presentation and framing. But what I meant to communicate with my previous comment #3 was that the theoretical significance of the mediation tests was not totally clear to me before I reached the discussion of this paper (around p. 10). Having read the whole manuscript, I think the mediation models help answer this question: to what extent are racial differences in concept relations attributable, specifically, to different political information ecosystems as opposed to other differences in their lived experiences? In other words, did Black and White respondents in this sample end up with different concept relations because they consume news with different political slants, or because their (non-political) lives and experiences are different? The "partial mediation" result here implies that political differences in media environments are one contributor to race differences in concept associations, but that they are not the whole story.

Is that what the reader should take away from these models? If so, could the authors do more to foreshadow that in the introduction and hypotheses?

I also suggested another analysis or set of statistics that the authors could present to cast light on this question: holding news sources constant, how similar/different were Black and White respondents in their concept maps? This is represented by the direct effects in the mediation models that the authors now present in the supplement – the effect of race (i.e., race differences in concept associations) tends to be significant even when controlling for media bias. So my question is answered, but this could perhaps get more explicit mention in the main text (if the editor agrees).

Meanwhile, the authors have also added information that answers additional questions:

Do concept relations differ more across political groups (as indexed by media consumption) than across racial groups? Evidence suggests that they do.

To what extent are the differences in concept associations across political groups also present across racial groups? This occurred for 18/44 concept pairs, suggesting that many (but of course not all) political differences in concept associations are associated with race.

Both of these things are good to know, but they seem to answer different questions than I assumed that the authors were trying to answer. Again, a more explicit presentation of these possibilities somewhere before the discussion would have helped me understand the implications of these results more immediately while reading.

EDITORIAL POLICIES

We ask that you ensure your manuscript complies with our editorial policies and reporting requirements.

To that end, we require revised manuscripts to be accompanied by two completed items: a reporting summary that collects information on study design and procedure, and an editorial policy checklist that verifies compliance with all required editorial policies.

- <https://www.nature.com/documents/nr-reporting-summary.zip>>Nature Research Reporting Summary
- <https://www.nature.com/documents/nr-editorial-policy-checklist.pdf>>Editorial Policy Checklist

All points on the policy checklist must be addressed. Your revised manuscript can only be sent back to the referees if these checklists are completed and uploaded with the revision.

Notes: If you have submitted a Stage 1 Registered Report, Review, Primer, Comment, or Perspective you do not need to submit these forms. If you have already submitted these forms, you may disregard this request.

** Visit Nature Research's author and referees' website at <http://www.nature.com/authors>>www.nature.com/authors for information about policies, services and author benefits**

If you experience problems in linking your ORCID, please contact the <http://platformsupport.nature.com/>>Platform Support Helpdesk.

Version 2:

Decision Letter:

Dear Dr Vargas,

Your manuscript titled "Identity and Informational Differences in Concept Relations for Social-Societal Institutions" has now been editorially evaluated, and I am delighted to say that we are happy, in principle, to publish a suitably revised version in Communications Psychology.

We therefore invite you to revise your paper one last time to address the remaining concerns of our reviewers and a list of editorial requests. At the same time we ask that you edit your manuscript to comply with our format requirements and to maximise the accessibility and therefore the impact of your work.

EDITORIAL REQUESTS:

SUBMISSION INFORMATION:

In order to accept your paper, we require the files listed at the end of the Editorial Requests Table; the list of required files is also available at <https://www.nature.com/documents/commsj-file-checklist.pdf> .

OPEN ACCESS:

Communications Psychology is a fully open access journal. Articles are made freely accessible on publication. For further information about article processing charges, open access funding, and advice and support from Nature Research, please visit <https://www.nature.com/commpsychol/open-access>

* **DATA AVAILABILITY:**

All Communications Psychology manuscripts must include a section titled "Data Availability" at the end of the Methods section. More information on this policy, is available in the Editorial Requests Table and at <http://www.nature.com/authors/policies/data/data-availability-statements-data-citations.pdf>

Link Redacted

Best regards,

Jennifer Bellingtier

Jennifer Bellingtier, PhD
Senior Editor
Communications Psychology

Yafeng Pan, PhD
Editorial Board Member
Communications Psychology
orcid.org/0000-0002-5633-8313

Nature Communications Psychology: Response-to-Reviewers

Reviewer #1 Comments and Responses:

0. This paper had participants who either identify as Black or White Americans rate the similarity of 16 concepts and identified the concept pairs on which the two groups systematically differ. Several of these racial group differences are then found to be partially mediated by the bias of the news sources the individual participants consume. The paper suggests that there exist systematic semantic differences between existing social groups, which are partly due to the groups' different (media) experiences. These results extend previous findings which have established similar differences between age, gender, and education groups. I suspect the results will be of interest to many readers both in social and cognitive psychology, as well as in sociology and media studies. The paper communicates an important message about the impact of media biases.

The data collection is sound, the figures provide face validity to the results, and the conducted analyses are sophisticated. There also lies the difficulty with the work. Its presentation in a rather short format makes it difficult to fully interpret and obscures some elements of the work. My main recommendations therefore reside at the level of the presentation, but I also provide several smaller comments. I do applaud the authors for making the data and code publicly available. It offsets some of the sparseness in the presentation.

Reply: We thank Reviewer #1 for appreciating the significance and broad appeal of our presented research. We also appreciate their clear, constructive, and helpful feedback on how to best portray our work. We have incorporated Reviewer #1's suggestions by clarifying the quality of conceptual information we believe to be measuring. Because we wanted to preserve the broad appeal of the work, we aimed to incorporate this change subtly as opposed to providing explanations of domain-specific terminology. In the following sections we list the reviewer's comments (in the form of a numbered list) and address each point individually.

1. Most people studying semantic cognition will not be surprised to learn that experiences shape semantic relationships. To appeal more to this audience, I recommend the authors take a little more space to explain the novelty of their contribution, which in my opinion lies in the demonstration of an effect of media consumption on semantic cognition and the demonstration of semantic differences between racial groups, which hasn't been shown as far as I know. To appeal to social psychologists more, the authors could make a stronger case that the relationships that are being studied are semantic ones and not opinions or values.

Reply: We appreciate that the reviewer provided specific suggestions on how to emphasize the novelty of the findings of the research. This is indeed a difficult balance given the space limitations of the journal. We have edited the Abstract and Introduction in several minor ways to clarify that we are looking at semantic judgements, and not exclusively opinions or values.

We have also added some additional language at the beginning of the Discussion that clarifies the quality of the conceptual information we are identifying. The added/modified text is as follows: "*Although people's values and opinions factor into their ratings of conceptual similarity, concept associations reflect the semantic structures that sculpt and filter an individual's view of their reality. In this way, a set of concept associations reflect a semantic knowledge that is inclusive of both what they know and how they feel.*"

In the closing paragraph of the Discussion section, we have included a few sentences emphasizing the strength of the media bias effect: "*Notably, the magnitude of the effect of news consumptions is considerably greater than that observed by racial identity.*"

2. I don't think social psychologists would be surprised to learn that the opinions of different racial groups differ on matters like voting, immigration or science. I don't think that the fact that you ask about the similarity of concepts would convince everyone that you are tapping into semantic knowledge if the latter is understood as "factual" knowledge.

Reply: We generally wanted to avoid drawing hard boundaries in our use of the label 'semantics'. Per our response to point 1 above, we provided a description of our stance on the quality of conceptual information measured in the study. We hope that this clarifies issue for readers with a social psychology background.

3. Related to this, the authors often use the term 'associability' to refer to the investigated relationships. In the semantic cognition literature, this refers to a rather specific type of semantic relation. I am not sure the authors are aware of this and want to argue that it is this specific type of semantic relation that is being studied/impacted?

Reply: We were previously unaware of a technical definition of 'associability' and had intended to use the term to describe relationships between conceptual nodes as described in Steyvers & Tenenbaum, 2005. We now clarify this in the opening paragraph of the Results by adding the following text and citation: "*The associability of a concept pair can be intuited as its weighted connection, that is, the magnitude of semantic relatedness (Steyvers & Tenenbaum, 2005)*"

4. In several places throughout the paper, the authors speak about the geometry of inter-concepts associations or concept graphs. If I understand the analyses correctly, the authors do not build geometrical or graph representations of the similarity data they obtained and compare these between groups, but rather conduct statistical inferences on transformed similarity data for individual concept pairs and use graphs (network representations) to illustrate the resulting differences simultaneously for the different pairs. Because of this, the emphasis on graphs and geometrical representations could perhaps be toned down a bit, because there is an extensive literature in which such representations are built and investigated, but that does not seem to be done here.

Reply: We generally agree with the author on this point. We have deemphasized our framing of geometries to a degree. The illustrations we generate have graphical properties (e.g., weights, connectivity) we present minimal results utilizing the graphs as data structures. Although we do not focus the current analyses on the graphs themselves, we hope to perform such analyses (e.g., weight thresholded *adjacency* lists) in future projects. We have opted to use the term 'concept map' where it would be more intuitive/appropriate.

5. It also raises the question why the authors decided to conduct their analyses at the level of concept pairs rather than at the level of all inter-concept relationships simultaneously? Perhaps they did not want to make any representational assumptions? In any case, I would appreciate an indication of the relative strengths and limitations of both approaches.

Reply: This is a point that we had considered early on and, in the end, the analysis approach was determined by necessity of the data set we could acquire. The final analyses contained 446 participants, and the number of concept pairs was 120. We feel we were insufficiently powered to compute a model that includes all 120 pair combinations as dependent measures. Although a multivariate approach would have highlighted collinearity in the concept-pairs, we opted to examine the pairs individually and apply a correction for family-wise error. As the reviewer mentioned, we didn't want to make any representational assumptions. We also feel our approach provided simplicity in interpretation by giving specificity to the

results of individual concept pairs. We have added the following language in the Methods section to justify our approach: *“Although a single multivariate model containing all 120 concept pairs could have been computed, due to statistical power constraints and our interest in individual concept pairs, regression modelling (with family-wise error correction) was computed on individual concept pairs.”*

6. Somewhat related: I hadn't come across the use of subgraphs to highlight edge differences between groups before. Is this a new visualization technique the authors came up with or has this been tried before? I ask, because it find it promising, but also wonder how good a job it does conveying the differences. For instance, to appreciate the differences between the groups in Figure 2 we need to integrate across a single graph representation for the aggregate data (panel a) and two separate graph representations (one for each racial group in panels c en d).

Reply: Using subgraphs to represent the edge differences is a visualization approach that we developed for this project. Given that the edge weights are a function of the similarity ratings, and the edges present within Figure 2 reflect those that differ by groups, it is impossible to create a graph highlighting differing edges that simultaneously allows for group-specific geometries to be visualized. That is, for one group we would expect two nodes to be closer together and for the other group we would expect them to be farther apart. Thus, we have opted to have panel 'a' as a summary of the differences depicting the pooled data across groups in a way that highlights the direction of differences for individual edges. All the edges in panel 'a' are represented in both panels 'c'/'d'. Our goal for these latter two panels is to visualize each group's graph in a way that prevents instances where edge differences are cancelled out across groups due to data pooling. That is, to show how each of the groups appears independent of one another. We have added additional texts in the Figure 2 caption that clarifies which edges are represented in panel 'c'/'d': *“Subgraphs of race-differing edges for Black and White participants, respectively, for the same edges and nodes as panel a.”*

7. I appreciate that the authors' main interest lies with racial group differences and how these are mediated by news consumption. I did find it interesting to see that news consumption appeared to yield more extensive effects than race, at least if one looks at the number of concept pairs that differed in the two comparisons (44 pairs vs 31 pairs). Do the authors consider this a substantial finding? For instance, can it be taken to indicate that experience is more important than group membership for semantic cognition? Intuitively, that would make sense to me.

Reply: This is a tricky issue as there are reasons that could lead to a stronger effect of news consumption than group identity, even if (in reality) group differences are a stronger effect. For example, (as raised by Reviewer #2's first comment below), sampling biases could lead to our sample not being completely representative of Black and White American perspectives, leading to a potentially weaker group effect.

So we have to balance making an inference from the sample that we have with the realities that the real relationships are more complicated. Nonetheless, it is true the magnitude of the effect size differences typically found for the differing news pairs is substantially larger than that for those identified by racial group in our data set. Additional text has been added to the final paragraph of the Discussion emphasizing this point.: *“The relative strength of the news consumption effect over racial identity is highlighted both by the number of concept pairs with differing edge lengths (44 compared to 31) and by the magnitude of rating differences observed along those differing concept pairs (Figure 2b; S11). This difference in impact suggests that informational ecosystems play a larger role than group membership for semantic cognition.”*

MINOR COMMENTS:

Minor 1. - p. 3: It was not immediate clear to me what you mean when you write that schemas should follow specific criteria for subsequent evaluation.

Reply: We have rewritten the sentence to be clearer. It now reads, “*It is theorized that news sources, through repeated coverage of issues surrounding specific concepts, communicate not only that concept schemas or frameworks are shared by a group of individuals, but also provide specific criteria for how concepts should be subsequently evaluated (Scheufele, 2004; Iyengar & Kinder, 2010).*”

Minor 2. - p. 3: The observation that selective consumption of media may enhance beliefs in conspiracy stories is a rather extreme example of the effects of media consumption on cognition. Can you provide a more moderate example as well?

Reply: The section has been rewritten to provide a clearer explanation of media effects and a less extreme example is provided. The section now reads as follows: “*Selective consumption of media messaging may prioritize the importance of certain topics over others and frame the perspectives people use to evaluate an issue or topic (e.g., viewing immigration as an economic threat or moral issue; Scheufele, 2004). In extreme cases selective media consumption may enhance beliefs in conspiratorial viewpoints (Romer & Jamieson, 2021).*”

Minor 3. - p. 4: Perhaps explicitly indicate that the graph in panel c is based on the average similarity data across the full sample.

Reply: The following text has been added to the Figure 1 caption: “*This graph is based on the average similarity data across the full sample of analyzed data (N = 446).*”

Minor 4. - p. 5: The notion of centrality is introduced in passing here. I recommend explaining further how it is operationalized. I take it the discussion of centrality is not meant to convey that the hubs differ between groups, but to signal which hubs are central in the relationships that differ between the groups? What can we learn from the observation of these specific hubs?

Reply: The second paragraph of the Results section has been modified to clarify the interpretation of centrality in the context of the graph of differing edges. It now reads as follows, “~~*A concept node’s centrality represents the magnitude that it serves as a hub (connecting other nodes) within a graph.*~~ *A concept node’s centrality in this context indicates how frequently it differs between groups, based on the number of unique edges it is connected to. Therefore, a concept’s centrality shows the extent of representational differences between groups.*”

Minor 5. - p. 6: The scale values along the horizontal axis of panel b in Figure 2 are missing.

Reply: Thank you for catching this. Panel d was previously occluding the scale on Panel b. This has now been corrected.

Minor 6. - p. 7: Perhaps it would be worth while spelling out explicitly that the relationship between entropy and news bias betas is not a necessary one. Obviously, for any predictor to have a significant effect there needs to be variability (entropy) in the variable that is being predicted, but not every predictor will be able to explain this variability. (Maybe this is obvious to most readers of the journal, but I had to think about this for a moment to ascertain myself that no spurious relationship was being reported.)

Reply: We have added the following text to the 5th paragraph of the Discussion section to more explicitly describe the relationship between news bias effects on concept pair entropy: “*Although our findings are consistent with past*

research suggesting that news bias effects influence conceptual representations, it is possible that the relationship between news bias effects and concept pair uncertainty could be mediated by unknown factors”

Minor 7. - p. 7: I found the passage that starts with “Of those 44...” and ends with “...White Americans, respectively.” difficult to follow, while it is very important for the paper. Could you please rephrase, elaborate, or provide an example in support?

Reply: This section has been revised for clarity and now reads as follows: “Among the 44 concept pairs that differed by news consumption, 18 also differed between Black and White Americans in a seemingly systematic way. Specifically, for this common subset of 18 group-differing edges, Left-leaning news consumers and Black Americans showed similar patterns of concept pair associations, whereas Right-leaning news consumers and White Americans displayed similar patterns.”

Minor 8. - p. 8: Which test was conducted to establish the bias difference in news consumption between Black and White participants? Note that often in the paper the word ‘across’ is used where I would use ‘between’ (as in the example of the bias test). To my non-native ear at least, the use of ‘between’ conveys better what is done.

Reply: Both mean differences with confidence intervals and permuted correlation analyses were used to establish the bias difference in news consumption between Black and White participants. This is explained in the “News bias mediated identity-based associations” section of the paper: “Although the total sample of participants tended to consume Left-leaning news (Figure 4a; mean bias = 2.29; SD = 1.23), there were significant differences in the direction of partisan news consumption across Black (mean bias = 2.04, 99% CI [1.84, 2.24]) and White participants (mean bias = 2.52, 99% CI [2.30, 2.74]; Figure 4b). A significant correlation between race and news bias consumption ($r = -0.195$, $p < 0.0001$) suggests that Black identifying participants consume more Left-leaning news and White identifying participants consume more Right-leaning news (although it should be noted that, in the aggregate, the magnitude of this difference is only about 0.5 points on the news bias rating scale).”

We have gone through the paper and changed “across” to “between” where appropriate. Thank you for pointing this out; we agree it makes the interpretations clearer.

Minor 9. - p. 8: Would it be possible to report the outcome of the mediation analyses somewhere, for instance in the supplementary info? This would make it easier to appreciate the difference between unmediated, partially mediated, and fully mediated race differences.

Reply: We now include a comprehensive table of the results of the mediation analysis in the Supplementary Information with the regression weights and FDR-corrected significance (SI3).

Minor 10. - p. 9: Figure 4d appears to use a form of color coding but this isn’t explained. For consistency, the authors could also consider using the same representation format as in Figure 2.

Reply: We thank the reviewer for catching our exclusion of an explanation of the color coding. It appears that in our figure generation code the color coding for the lines in Panel 4 were inverted. The color coding in Figure 4 communicates different information from that displayed in Figure 2. The color coding in Figure 2 refers to the direction of race-differing edges while the color coding in Figure 4 denotes news mediated edges. We have now corrected Figure 4 and added an explanation of the color coding to the caption: “(d) Subgraph of all race-differing edges excluding the 3 edges that were completely mediated by news consumption. Purple lines denote edges that are partially mediated by news consumption but also have a direct effect of race, that is, they are partially mediated by news consumption. Grey lines denote race differing edges that did not differ by news consumption.”

Minor 11. - p. 10: The concepts police, religion, and science are here identified as hubs for the mediated news effects, but I don't think this has been established yet in the results section (other than shown in Figure 4).

Reply: The centrality metrics have been added to the Figure 4 caption.

Minor 12. - p. 11: Please elaborate on the significance of the following sentence: Measures of news consumption do not describe views on dimensions used to characterize political conflict including limited government and cultural conservatism.

Reply: The sentence has been modified to provide clarity and now reads as follows: "Measures of news consumption do not describe the specific quality of an individual's views such as beliefs on *limited government* and *cultural conservatism* (Bartels, 2018)."

Minor 13. - p. 12: What was the rationale behind the choice for the 16 concepts? The title hints at the relevance of social-societal institutions, but this isn't further explained. A word on presentation order of concept pairs and words within pairs during the pairwise rating task would be appreciated as well. I applaud the authors for computing the reliability of the similarity ratings. This is a pre-requisite for its visual representation.

Reply: We selected this set of socioenvironmental concepts because there is empirical evidence in the attitudes literature supporting that differences exist for these concepts along Black and White racial identity boundaries. We imply this in the opening of the final paragraph of the Introduction: "*As groups, White and Black identifying Americans often have different attitudes and beliefs on issues like law enforcement (Pew Research Center, 2020), healthcare (Armstrong et al., 2007; LaVeist et al., 2000; Pew Research Center, 2019), religion (Pew Research Center, 2021), and political leaning (Gilens, 2023).*" The concept *Firefighter* was included because we wanted a concept that we did not expect to differ across groups.

We have added a few sentences in the Methods section clarifying our concept selections: "*These socioenvironmental concepts were selected based on previous research on attitude that showed discrepancies between Black and White Americans. Firefighters was an exception in that it was selected because there were no expected identity-based differences.*"

The following sentence was added to the end of the first paragraph of the Experimental procedure section of the Methods: "*The presentation order of the concept pairs was randomized, however, their left-right positioning was not. The mean probability of left positioning across the 16 concepts was 50% with a range of 0-100%. Although, we do not suspect the left-right positioning of the concepts to influence ratings, future implementations of this task will aim to balance this.*"

Minor 14. - p. 12: I would appreciate it if the outcome of the general linear model would be reported somewhere, for instance in the supplementary info. The model includes mean news consumption bias, but I don't think we learn anything about this variable's contribution in the model.

Reply: We have provided a table in the Supplementary Information (SI5) that lists the full model, including regression weights for each predictor across all 120 concept pairs.

Minor 15. - p. 13: Is the formula for Shannon's entropy traditionally not preceded by a minus sign?

Reply: Excellent catch. This was a typo. We have now corrected the formula to include the minus sign.

Reviewer #2 Comments and Responses:

0. This paper uses an interesting and novel (at least to me) approach to explore how associations between concepts vary across groups. The argument—and the evidence—suggests that groups of people may develop different “concept maps” of the world. Different groups not only evaluate different individual people, ideas, and things differently, but also perceive different relations among those objects. These group differences seem to be partly attributable to (or at least associated with) group members’ tendency to reside in different “information ecosystems.” Such as, in this case, relatively liberal versus conservative media environments in the U.S.

I think this question of how different groups perceive and map reality differently is very interesting. It reminds me of research on political ideology, which has long grappled with similar questions of “what goes with what” and for whom, but the authors present a method that could be adapted to explore relations between an infinite variety of idea elements that may or may not be political. I think their data would be interesting to many readers.

Reply: We thank Reviewer #2 for appreciating the novelty and broad appeal of our presented research. We also appreciate their clear, constructive, and helpful feedback on how to best clarify our work both in terms of the methods used and the possible limitations of the sample. We have incorporated Reviewer #2’s suggestions by adding a dedicated limitation section describing how our sample may not be representative of the general populations and the steps we took to mitigate these possible differences. Additionally, we have provided a more comprehensive description of the results in the supplementary information and reworded and elaborated on our description of some of the analyses for greater clarity. In the following sections we list the reviewer’s comments (in the form of a numbered list) and address each point individually.

1. Because the work is at least partly descriptive and not experimental, the sample coming from MTurk is a potentially significant limitation. I would like the paper to include more discussion about how representative this sample is of the intended target population(s). This is especially important given the authors’ interest in subgroups; the question is not just how MTurkers differ from the U.S. population as a whole, but how White and Black MTurkers differ from White and Black Americans, and whether and how their concept maps would likely differ as a result. I defer to the editor on the importance of representativeness for publication, but I do think this limitation merits more page space, with a clearer presentation of what is safe vs not safe to conclude from this sample.

Reply: This is a reasonable point that reflects a concern with all such online based samples. Nonetheless it is important for the naïve reader to understand. We have thus added the following *Limitations* section at the end of the Discussion that highlights concerns of representativeness among crowdsourced samples: “**Limitations.** *Although participants recruited through crowdsourcing platforms are often more representative than locally recruited college samples, they are often less diverse than the general US population and suffer from concerns about response quality (Fowler et al., 2023). Crowdsourced participants tend to be more educated, younger, and predominantly White identifying relative to the general population (Pew Research Center, 2016). Moreover, crowdsourced participants tend to generally be politically left leaning, as we have observed in the distribution of news bias scores in our sample. Yet, the values and personality traits typically held by conservative and liberal individuals recruited through crowdsourcing tend to reflect the general population (Clifford et al., 2015). To mitigate issues of representativeness, selective recruitment was implemented to target groups of interest to allow for large samples of participants to be recruited within each group. Moreover, participants were recruited through Cloud Research Connect or through Cloud Research’s TurkPrime Toolkit which tend to have higher response quality relative to recruitment through MTurk directly (Douglas et al., 2023). Additionally, our sample of Black and White participants show trends in political leaning consistent with past research (Gilens, 2023). In summary, although steps were taken to mitigate*

unrepresentativeness within the sample, it is possible the results presented here deviate from the general population.”

2. The authors mention that they analyzed “various facets of identity (including race, gender, income, and age).” I realize space here is limited, but even if the authors don’t have time to cover what all of the group-related differences were, I would be curious to know *whether* concept maps differed depending on age, gender, or other group identities.

Reply: We have included a comprehensive table of the model results in the supplementary materials that shows how facets of identity predict each of the 120 concept pairs (SI5). This table includes regression weights and FDR-corrected significance for all model predictors (i.e., race, gender, income, and news consumption) for the 120 concept pairs.

3.1. The discussion talks about this some already (p. 10), but I would like to see a bit more attention (in the results or intro) to what “partial mediation” (or even just controlling for media environment) means in this context, with more done to unpack it in the quantitative analyses. I think this is a really vital question for this paper – Among conservative news consumers (or among liberal news consumers), are Black and White respondents very similar in their concept maps? Or still pretty different? Answering this question could help clarify the extent to which group differences in concept maps are the product of media ecosystems versus lived experience (or any other group differences not associated with the political slant of the news people watch).

Reply: We have modified the language when introducing the idea of partial mediation when we list the hypotheses in the Introduction. The hypothesis now reads as follows: “(c) that informational ecosystems will partially account for (i.e., partially mediate) identity-based differences in concept graphs.”

Additionally, we have included a schematic of the mediation analysis (see SI2, and a comprehensive table of the mediation model results (including the model direct and indirect effects) in the Supplementary Information the beta weights and FDR-corrected significance for all model predictors (see SI3).

Regarding the question of: “**Among conservative news consumers (or among liberal news consumers), are Black and White respondents very similar in their concept maps?**”, we provide an explanation of this combination of race and partisan news consumption in the Discussion section, however, as Reviewer #1 pointed out the section where this is explained was written in a way that was unclear. We have since clarified this section and it now reads as follows: “*Among the 44 concept pairs that differed by news consumption, 18 also differed between Black and White Americans in a seemingly systematic way. Specifically, for this common subset of 18 group-differing edges, Left-leaning news consumers and Black Americans showed similar patterns of concept pair associations, whereas Right-leaning news consumers and White Americans displayed similar patterns.*”

3.2. Attending to other group differences could also help answer this question. For example, did any other groups differ from one another in their political orientation as strongly as White and Black respondents did, and did they show similar patterns of mediation? Did other groups with identical political views end up with identical concept maps?

Reply: All the other identity measures we obtained including Age, Income, and Gender did not show reliable differences among the concept pairs. This is a necessary condition for implementation of the mediation analysis of News Consumption Bias. Because of this, we hadn’t previously thought to look to what extent News Consumption Bias differed for these variables, however, we also feel this could potentially be interesting to examine these relationships. We have included correlation statistics in the Supplementary Information (SI4) reporting the correlations between news consumption and age, income, and gender: “*Age was modestly correlated with Right-*

leaning news consumption, $r = 0.1$ ($p = 0.02$). News consumption was not correlated with either Income ($r = -0.04$, *n.s.*) or Gender ($r = 0.04$, *n.s.*.)”

Some of the methods and results could be clarified for readers who (like me) have limited knowledge of the network analyses that the authors employ. Specifically:

4.1. The authors provide a lengthy and helpful definition of “edge length,” and then provide figures with both edge length and edge width, which, to my untrained eye, appear to quantify basically the same thing (associations between constructs) in opposite directions. What is the difference between length and width?

Reply: An edge’s length and width reflect similar information, that is, the strength of association between two nodes. We chose to have this redundancy to emphasize this relationship. The description of the edges in the Figure 2 caption has been revised to make this redundancy clearer: “*Concept edges are represented by the edge width/length (wider/closer = stronger association; thinner/farther = weaker association).*”

4.2. What is “Shannon entropy”?

Reply: Shannon’s entropy is a measure of uncertainty or randomness in data and has a long history of being used in cognitive and computer science. Intuitively it is a measure of the average uncertainty for a given variable as a measure of its potential states. The more possible states a signal can be in, the higher its entropy. In our case, this reflected the range of possible similarity rankings an individual concept pair could take across participant ratings. That is, if we were to pull the similarity ratings for a random pair of concepts, entropy would tell us how certain (or uncertain) we should be about what that similarity rating would be. If everyone in the sample rated a concept pair with the same value (say 7 on our scale), then we have a high degree of certainty about the state of similarity of that concept pair.

Given the ubiquitous use of Shannon’s entropy measure and the space limitations of the journal we have opted to not define it in the text.

Minor concerns:

5.1. I’m not sure whether “bias” is the right word to use for the overall ideological slant of news media, given that neutrality may sometimes indicate bias too. Maybe this is just the term that AllSides gives its variable, though?

Reply: The reviewer correctly ascertained that the label given to news sources by AllSides media are discussed in terms of “bias”. To be consistent with the labels provided with the source material we would prefer to retain the use of bias.

5.2. On page 8, the authors write “the bias of news sources that an individual gets their information from does appear to influence expression of group differences in concept relationships.” This is probably too strong a causal claim.

Reply: We agree that the mentioned statement is a bit strong. We have revised the sentence accordingly: “*Thus, the bias of news sources that an individual gets their information from may influence the expression of group differences in concept relationships.*”

Nature Communications Psychology: Response-to-Reviewers – Round 2

Reviewer #1 Comments and Responses:

0. I was Reviewer #1 of the original manuscript. I appreciate the authors' response to my comments and find the revised manuscript much clearer than the original. I have a few outstanding comments, but these can probably be addressed through minor revisions.

Reply: We thank Reviewer #1 for their past and present clear and constructive feedback. Their diligence and clarity both broaden the appeal of this research to a broader audience and making the overall conclusions more concise.

1. On p. 3, the introduction of semantic relationships feels rather abrupt. The preceding opening paragraph talks about ideas, values, and goals, which are typically the focus of social psychology studies in which racial groups have already been shown to have different attitudes toward social-societal institutions. The second paragraph does not explicitly indicate whether the authors think the semantic relationships they study should be considered different from these attitudes or not. In the discussion (p. 11) this becomes clearer, but I would recommend communicating this earlier because it signals the novelty of the contribution. At some points in the manuscript, the authors do not seem to make a distinction at all. On p. 10, for instance, they speak about their results in terms of “racial differences in attitudes on socioenvironmental concepts”.

Reply: We thank the reviewer for this insight to help the intellectual flow of the paper. Part of the difficulty of writing for journals with tight word lengths is balancing completeness with conciseness. These suggestions are helpful in trying to achieve this balance. We have revised part of the first paragraph of the Introduction to better introduce our framing of semantic relations: *“Consequently, group frameworks represent a collectively shared identity, defined not only by a common set of ideas, values, and goals, but also by a collective semantic knowledge that serves as a lens through which the environment is given meaning.”*

Additionally, we have corrected our contextualization of differences in attitudes as differences in representations. The instance pointed out by the reviewer in the Discussion (on pg10) now reads as follows: *“One possible explanation for racial differences in the representation of socioenvironmental concepts may be the historical and empirical disparities in how Black and White Americans have experienced public-serving institutions.”*

We looked through the rest of the manuscript to catch additional mischaracterizations or gaps in our logical flow.

2. I still feel talking about concept graphs and concept maps is unnecessary and somewhat obscures the fact that the analyses are performed at the level of individual concept pairs instead of across the entire stimulus domain. I would find the text easier to digest if it were just to speak about inter-concept semantic associations. For instance, in the abstract one could eliminate the graph language without affecting the content: “Here we set out to see how inter-concept semantic associations differ between social groups, based on racial identity, and how this effect is mediated by information

ecosystems, contextualized as news sources.” The graphs could be identified in the method section as a way of visualizing the results.

Reply: We can see where this concern might arise, but we think it may come from a slight misunderstanding of the general graph theoretic approach. First, edge-based analysis is a common approach in graph/network theory. Many analytical techniques used in graph theory involve iterative analysis on individual edges. For example, in neuroscience a functional connectome (the graph of functional relationships between brain areas) is established by iteratively measuring the pairwise correlations of functional activity between pairs of brain regions. In our work here, while some of the results are interpreted on the level of individual concept-pairs, the key analyses, both the glm and mediation model, are conducted on the edge level rather than raw ratings. This makes the relationships between individual concepts (nodes) the important factor, which by definition makes this analysis focused on the graphical relations between nodes (concepts). Second, in the introduction we define collective knowledge structures as a set of common associations; we feel that presenting the results as a collective graph structure is consistent with the framework. Finally, we do run analyses on the graph level. This includes an analysis of centrality reflecting which concepts serve as hubs of the differences across groups. So while the edge-wise analyses set up and define our graphs and subgraphs of concept relations, we analyze and interpret the results at the graph level.

However, we realize that we need to introduce this framing at the outset. In the abstract we have revised the section identified by the reviewer as follows: *“Here we set out to measure how a set of inter-concept semantic associations, comprising what we refer to as a concept graph, covaries between established social groups, based on racial identity, and how this effect is mediated by information ecosystems, contextualized as news sources.”*

MINOR COMMENTS:

Minor 1. -p. 5: Is there a straightforward way to interpret the values for centrality? Does the centrality value perhaps reflect the proportion of pairs that feature the concept which show a statistical group difference?

Reply: The reviewer is correct in their intuition of centrality. The centrality (i.e., normalized degree centrality) of a node is computed as the number of connected nodes divided by the total number of nodes minus 1. We thank the reviewer for articulating this concisely; their language is now reflected in the manuscript in the Results section titled **‘Differences in concept geometry across racial identities.’** where centrality is introduced.

Additionally, there was a visual bug in the graphing code that slightly altered the centrality values for Figure 4 panel ‘d’. This did not change which nodes were most central or the overall pattern of relations among the nodes. The panel ‘d’ has been updated and the centrality values have been corrected in the figure caption.

Minor 2. - p. 8: How are left- and right-leaning news consumers defined? Is it based on the center scale value or on a median split?

Reply: Left and right news consumers were defined as those whose mean news bias was above or below 3 (i.e., Center). Per the reviewer’s later comment, a note of this has been added to SI1 defining Left-Right

news consumers: “*Left and right news consumers were defined as those whose mean news bias was above or below 3 (i.e., Center)*”

Minor 3. - p. 9: ‘firefighter’ was included because the authors expected no identity-based differences for this concept, yet it did show race differences. Personally, I am not surprised that it shows a group difference, but perhaps the authors want to say something about it, since its inclusion seems to be based on methodological considerations?

Reply: This is a good point. Because of space limitations we have added our interpretations to SI5: “*Although no differences were initially expected for the concept firefighter, several differences emerged. Notably, firefighter-conservatives and police-firefighter associations differed by race, while firefighter-liberal varied by news consumption. One explanation could be that White and Right-leaning participants may associate firefighters and police as belonging to the same group (i.e., public servants), whereas Black and Left-leaning participants may view them as distinct entities. This is further supported by the observation that police was generally associated with conservatism across all groups.*”

Minor 4. - p. 11: “The effects of partisan news consumption in our study were also proportional with the amount of uncertainty with which people consider socioenvironmental concepts.” I believe ‘pairs’ should be added to this sentence, as the relationship pertains the concept pairs, not the individual concepts. Similar for the sentence later in the paragraph that reads “increased uncertainty in socioenvironmental concepts”.

Reply: Thank you for spotting these omissions. This has now been corrected.

Minor 5. - p. 13: SIX should read SI5.

Reply: Thank you for spotting this typo. This has now been corrected.

Minor 6. - SI1: “Associativity ratings are pooled across...” Here the authors could also add how they determined left- vs right-leaning news consumers (see earlier comment).

Reply: As mentioned in our response to the earlier comment, we have included the explanation in this location.

Minor 7. - SI4: Perhaps the authors could also provide an indication of the relationship between race and gender.

Reply: We have included the gender/race counts in SI4: “*Of the 446 participants, 114 identified as White and Male, 119 as White and Female, 111 as Black and Male, and 102 as Black and Female.*”

Reviewer #2 Comments and Responses:

0. I appreciate the substantial revisions that the authors have made to incorporate feedback from me and the rest of the review team. I think the manuscript is significantly improved. I have just a couple remaining suggestions.

Reply: We thank Reviewer #2 for their past and present clear and constructive feedback. Their insights have helped clarify our explanation of many of the key analyses and helped highlight the conclusions of the paper.

1. First, I was \ glad to see the new “limitations” paragraph about the sample. That said, I think it would be helpful if the authors could talk more specifically, however briefly, about the racial subgroups that are the focus of their analyses. It is good to note how MTurkers tend to differ from Americans in general, but what I was particularly curious about here was how their Black respondents differed from Black Americans in general and how their White respondents differed from White Americans (e.g., in income, education, political attitudes). If this information is available to the authors, it would provide useful context and help inform readers as to the likely generalizability of the results.

Reply: We appreciate the reviewer’s cognizance of the space constraints we have to juggle. To address this comment we now include in SI4 a comparison of how our other variables of interest including income and age with the current United States census statistics. Sadly, because of how education was queried it wasn’t possible to accurately extract that information from this sample, but this has been corrected for future studies. Moreover, in SI4 we reiterate that our findings are consistent with past literature regarding the relationship between race and political affiliation (Gilien, 2023). The added section in the SI is as follows: *“The mean age of the analyzed sample was 40.56 years (SD = 12.43). For Black and White identifying participants, the mean ages were 37.99 and 42.91, respectively, compared to the national averages of 36.3 and 44.2 (US Census, 2023). In this study, income was measured using bracket ranges, so direct comparison to national averages was not possible. To approximate the sample’s income, counts were converted to the midpoint of each range (e.g., \$12,501–\$22,500 was recorded as \$17,500), or to the exact value for open-ended responses (e.g., 'below \$12,500' was recorded as \$12,500). Based on this method, Black and White participants had mean incomes of \$42,142.39 and \$44,285.65, respectively, compared to the national averages of \$32,360 and \$50,675 for individual earners (US Census, 2023). The association in our sample between Black identity and Left-leaning political affiliation, and White identity and Right-leaning political affiliation, aligns with trends found in previous research (Gilens, 2023).”*

2. The sentence “our sample of Black and White participants show trends in political leaning consistent with past research (Gilens, 2023)” is an example of this type of information - this was helpful. I assume that by this the authors meant that the Black participants were generally more liberal than the White participants.

Reply: The reviewer’s intuition was correct. This sentence has been revised to make this point more explicit. Given that this point is mentioned now in SI4 with the other details of the participants, this sentence has now been removed from the limitations section.

3. As the authors point out, this sample limitation is far from unusual. But given the goal of the project – to describe differences in how Black and White Americans perceive the world – I think it

is unusually important in this study to make clear precisely whose perceptions are being described.

Second, I still think the authors could go a bit further to contextualize and explain their mediation results. This is a relatively minor quibble about presentation and framing. But what I meant to communicate with my previous comment #3 was that the theoretical significance of the mediation tests was not totally clear to me before I reached the discussion of this paper (around p. 10). Having read the whole manuscript, I think the mediation models help answer this question: to what extent are racial differences in concept relations attributable, specifically, to different political information ecosystems as opposed to other differences in their lived experiences? In other words, did Black and White respondents in this sample end up with different concept relations because they consume news with different political slants, or because their (non-political) lives and experiences are different? The “partial mediation” result here implies that political differences in media environments are one contributor to race differences in concept associations, but that they are not the whole story.

Is that what the reader should take away from these models? If so, could the authors do more to foreshadow that in the introduction and hypotheses?

Reply: The reviewer interpreted the goal of the mediation analysis perfectly. To make this point clearer, within the space constraints we are up against, we have borrowed some of the reviewer’s language to present the framing of this question in the last paragraph of the Introduction: *“That is, to what extent are group differences in concept associations attributable to related information consumption rather than possibly lived experiences?”*

4. I also suggested another analysis or set of statistics that the authors could present to cast light on this question: holding news sources constant, how similar/different were Black and White respondents in their concept maps? This is represented by the direct effects in the mediation models that the authors now present in the supplement – the effect of race (i.e., race differences in concept associations) tends to be significant even when controlling for media bias. So my question is answered, but this could perhaps get more explicit mention in the main text (if the editor agrees).

Reply: As mentioned in our response to the previous comment, the reviewer is correctly interpreting exactly what the mediation analysis aimed to address. We have edited the summary sentence of the ‘**News bias mediates identity-based associations**’ section in the Results: *“Moreover, while news consumption partly explains racial differences in the overlapping 18 concept pairs, these differences persist even after accounting for news consumption.”*

Also, to emphasize this key finding in the paper, we have more explicitly mentioned this point in the first paragraph of the Discussion section when summarizing the findings.

5. Meanwhile, the authors have also added information that answers additional questions: Do concept relations differ more across political groups (as indexed by media consumption) than across racial groups? Evidence suggests that they do.

To what extent are the differences in concept associations across political groups also present across racial groups? This occurred for 18/44 concept pairs, suggesting that many (but of course not all) political differences in concept associations are associated with race.

Both of these things are good to know, but they seem to answer different questions than I assumed that the authors were trying to answer. Again, a more explicit presentation of these possibilities somewhere before the discussion would have helped me understand the implications of these results more immediately while reading.

Reply: This builds nicely off of the responses and edits to the prior two comments. We have edited the summary sentence of the '**News bias mediates identity-based associations**' section in the Results: "*Concept relations differed more across Left and Right news consumers than across Black and White racial groups, as evident by the number of concept pairs that show differences (i.e., 44 vs 31).*"